# CircuitTuning: Improving Math Reasoning in LLMs via Targeted Sub-Network Updates

## Abstract

Prior studies investigating the internal workings of LLMs have uncovered sparse subnetworks, often referred to as circuits, that are responsible for performing specific tasks. Additionally, it has been shown that model performance improvement through fine-tuning often results from the strengthening of existing circuits in the model. Taken together, these findings suggest the possibility of intervening directly on such circuits to make precise, task-targeted updates. Motivated by these findings, we propose a novel method called *CircuitTuning* which identifies pivotal tokens from model reasoning traces as well as model components responsible for the desired task, and updates only those components. Applied to mathematical reasoning, it improves accuracy by up to +11.4% across multiple models while modifying as little as 1.59% of model components, with minimal impact on other abilities as measured by MMLU, TriviaQA, and TruthfulQA. These results demonstrate that targeted capabilities can be reliably enhanced by selectively updating a sparse set of model components.

## 1 Introduction

Large language models (LLMs) have demonstrated impressive general-purpose reasoning abilities, yet they continue to struggle with mathematical reasoning tasks, where even small logical errors can derail problem-solving (Shojaee et al., 2025; Marjanović et al., 2025; Ballon et al., 2025). Existing works have attempted to improve math reasoning through various prompting and fine-tuning strategies, which have led to modest gains (Wang et al., 2022b; Chen et al., 2022; Lewkowycz et al., 2022; Lightman et al., 2023). In this work, we propose an alternative approach that leverages insights from mechanistic interpretability to achieve more targeted improvements.

Recent progress in mechanistic interpretability has revealed that model behavior is often governed by sparse subnetworks, or circuits, consisting of attention heads and MLP neurons that jointly implement specific capabilities (Wang et al., 2022a; Hanna et al., 2023; Merullo et al., 2023; Prakash et al., 2024; Marks et al., 2025). Jain et al. (2023); Prakash et al. (2024); Chhabra et al. (2025) shows that fine-tuning frequently strengthens these existing circuits rather than creating entirely new mechanisms. Additionally, Rai et al. (2025); Ortu et al. (2024) suggest that there is a competition among circuits within a model's internal computation, where some circuits contribute to correct reasoning while others introduce noise. Together, these findings suggest that targeted interventions on circuits could enable precise updates that enhance specific skills while minimizing unrelated disruption.

In this work, we introduce **CircuitTuning**, a mechanistically informed fine-tuning method that performs sparse, targeted updates to improve LLM reasoning. CircuitTuning operates in three stages: (i) generating reasoning traces to identify pivotal tokens where incorrect solutions diverge from correct ones, (ii) localizing the attention heads and MLP neurons that promote correct reasoning paths, and (iii) applying gradient updates exclusively to those components. By amplifying the contribution of the circuits most responsible for correct reasoning, CircuitTuning strengthens mathematical reasoning ability while leaving unrelated skills largely intact.

Applied to the GSM-Symbolic benchmark Mirzadeh et al. (2025), CircuitTuning yields accuracy improvements of up to +11.4% when tested across multiple model families, while modifying as little as 1.59% of components (i.e. attention heads and MLP neurons). Importantly, these gains come with minimal degradation on general benchmarks including MMLU, TriviaQA, and TruthfulQA,

underscoring that targeted improvements can be achieved without compromising broad capabilities (Hendrycks et al., 2021; Joshi et al., 2017; Lin et al., 2022).

Our results demonstrate that LLM skills can be selectively enhanced by updating only the sparse subnetwork that implements them. Beyond mathematical reasoning, this highlights a broader principle: mechanistically guided, sparse updates provide a pathway toward interpretable model adaptation.

## 2 RELATED WORKS

### 2.1 MECHANISTIC INTERPRETABILITY IN LLMS

The field of mechanistic interpretability seeks to reverse-engineer the internal computations of deep neural networks (Olah et al., 2020; Mueller et al., 2024; Saphra & Wiegreffe, 2024). A prominent line of work focuses on uncovering *circuits*, sparse sets of attention heads and MLP neurons that collectively drive specific model behaviors, such as indirect object identification, greater-than, and entity tracking Wang et al. (2022a); Hanna et al. (2023); Prakash et al. (2024). Recent research has also extended this perspective to the sparse feature space, identifying and editing interpretable circuits that govern feature-level interactions Marks et al. (2025); Ameisen et al. (2025).

A recurring theme across this line of work is that LM behavior is not uniformly distributed across parameters, but rather localized within a relatively small subset of components. Jain et al. (2023); Prakash et al. (2024); Chhabra et al. (2025) show that fine-tuning often strengthens existing circuits rather than creating entirely new mechanisms, while (Merullo et al., 2023) highlights how subcircuits are reused across different tasks. These findings motivate our approach of selectively amplifying the circuits responsible for the target task, while minimizing disruption to unrelated capabilities.

### 2.2 MATHEMATICAL REASONING WITH LLMS

Improving mathematical reasoning in LLMs has been a central challenge, as even minor logical mistakes can derail otherwise promising problem-solving attempts (Wang et al., 2025). A line of research has focused on prompting strategies, such as chain-of-thought prompting, self-consistency, and program-of-thoughts prompting, which encourage models to externalize intermediate steps and thereby improve reliability (Wang et al., 2022b; Chen et al., 2022; Lightman et al., 2023). Another line of work investigates fine-tuning techniques, including supervised fine-tuning on reasoning traces or parameter-efficient approaches like LoRA, which can adapt models toward stronger mathematical reasoning (Lewkowycz et al., 2022).

Complementary to behavioral approaches, recent research has also examined the internal mechanisms of LLMs to better understand their mathematical reasoning capabilities. For example, Ye et al. (2024) analyzed the internal activations of a transformer model trained from scratch on a math reasoning dataset, using probes to uncover mechanisms underlying the reasoning ability. Similarly, Sun et al. (2025b) trained probes to predict the correctness of outputs in 3-digit addition, showing strong generalization to addition-only GSM8K problems. By leveraging these probes, they selectively re-prompted erroneous reasoning steps, thereby improving task accuracy. A closely related study, Sun et al. (2025a), introduced ThinkEdit, which identifies attention heads responsible for short reasoning traces and updates their weights to extend these traces, ultimately enhancing model performance. Building on this line of work, we show that localization-informed model update can not only affect reasoning trace length but also strengthen mathematical capabilities, enabling targeted interventions to improve overall performance.

## 3 CIRCUITTUNING

### 3.1 METHOD OVERVIEW

We propose a novel technique, called *CircuitTuning*, to improve the mathematical reasoning capabilities of an LM, without affecting other abilities. The underlying premise of this method relies on two empirical insights from the mechanistic interpretability literature: 1) Specific tasks in LM are often executed by a sparse subnetwork, which gets augmented during fine-tuning, leading to model performance improvement (Jain et al., 2023; Prakash et al., 2024; Chhabra et al., 2025). 2) There

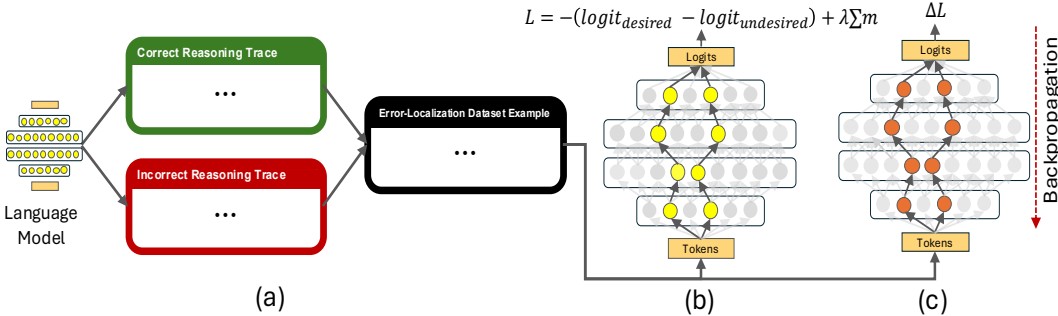

$$L = -(logit_{desired} - logit_{undesired}) + \lambda \sum m \quad \Delta L$$

(a)          (b)          (c)

Figure 1: **Overview of CircuitTuning**: (a) *Token Localization*: For a given problem, we generate both correct and incorrect reasoning traces and identify the pivotal token where the incorrect trace diverges from the correct one. The intervention point is chosen as the token immediately preceding this divergence. (b) *Model Component Localization*: Using the Error-Localization dataset constructed from these reasoning trace pairs, we apply Desiderata-based Component Masking (DCM) to learn a sparse binary mask over attention heads and MLP neurons. This identifies the subset of components that most strongly promote the desired token. (c) *Model Update*: Gradient updates are then applied exclusively to the localized components, amplifying constructive computations while leaving the rest of the network unchanged.

is a competition among various mechanisms within an LM's internal computation, some of which are sound for the given task, while others are introducing noise, as suggested in Rai et al. (2025); Ortu et al. (2024). In addition to existing works in the literature indicating such a phenomenon inside LM's internal computation, a good behavioral performance of the models that we investigate suggests that the models have a decent idea of solving the math reasoning tasks; however, on certain tasks, they deviate towards incorrect reasoning, which leads to an incorrect final answer. To overcome this shortcoming, CircuitTuning amplifies the signal from model components that are constructively generating the correct response. The technique consists of three steps: 1) Generation of Error-Localization dataset, 2) Training binary mask to localize constructive model components, and 3) Updating only those model components using a few gradient update steps. The following subsections describe each step in more detail, including its role and the procedure.

### 3.1.1 LOCALIZING REASONING ERRORS

The first step of CircuitTuning is to identify the point in the reasoning trace where the model begins to deviate toward an incorrect answer, as illustrated in Figure 1(a). Prior work on circuit discovery has primarily examined tasks where the output is produced *in a single forward pass*, such as indirect object identification, entity tracking, and greater-than comparison (Wang et al., 2022a; Prakash et al., 2024; Hanna et al., 2023), making it natural to apply circuit discovery methods at the final token position. In contrast, mathematical reasoning involves multi-step computations, and it is less clear at which point to apply the circuit discovery algorithm to uncover the circuit that can be enhanced to improve overall reasoning ability. To address this, we begin circuit localization by identifying the token in the reasoning trace where an intervention should occur. Specifically, for a GSM-Symbolic instance where the model produces an incorrect solution through greedy sampling, we aim to locate the first token in the reasoning trace that drives the model toward this error. We refer to this token as the *pivotal token*. Our intervention then targets the token immediately preceding it, to discourage the model from generating the pivotal token. We refer to this intervention point, where we amplify signals from constructive model components, as the *intervention token*.

For each GSM-Symbolic instance, we first sample a reasoning trace and final answer using greedy decoding. If the answer is incorrect, we generate an alternative reasoning trace that leads to the correct answer via non-greedy decoding. Conversely, if greedy decoding produces the correct answer, we instead generate an incorrect reasoning trace using non-greedy decoding. With this paired set of reasoning traces, we then apply one of the following methods to identify the intervention token.

**Correct Reasoning Trace**

**Q:** There are 67 fourth-graders at Sunny Hill School. 38 of them are girls. On Thursday, 6 fourth-grade girls and 6 fourth-grade boys were absent. How many fourth grade boys were at Sunny Hill School on Thursday?
**A:** Let's think step by step. There are 67 fourth graders in total. 38 are girls, so 67 - 38 = 29 are boys. 6 girls were absent, but the question asks about the number of boys present. So 29 - 6 = 23 boys were at school on Thursday. The final answer is 23.

**Incorrect Reasoning Trace**

**Q:** There are 67 fourth-graders at Sunny Hill School. 38 of them are girls. On Thursday, 6 fourth-grade girls and 6 fourth-grade boys were absent. How many fourth grade boys were at Sunny Hill School on Thursday?
**A:** Let's think step by step. There are 67 fourth graders total. 38 are girls. That means 67 - 38 = 29 are boys. Thursday 6 girls were absent, but no information about how many boys were absent. So, all 29 boys were at school.

☐ Pivotal token identified by Prefix method ☐ Pivotal token identified by Branching method

Figure 2: Example of a GSM-Symbolic math word problem showing both a correct and an incorrect reasoning trace produced by the `Gemma-2-9b-Instruct` model. The correct trace (top) is obtained through greedy decoding, while the incorrect trace (bottom) is produced by non-greedy sampling.

**Prefix Method:** Given a pair of reasoning traces, this method identifies the first token that is not shared between them as the pivotal token. For instance, in the Figure 2, the first uncommon token between both the reasoning traces is the "," and "." tokens. Consequently, its prior token, i.e. "*girls*", becomes the intervention token.

Although efficient, this method can sometimes identify suboptimal pivotal and intervention tokens. For example, in Figure 2, the first differing tokens in the two traces are "," and ".". However, these tokens are not the decisive points that steer the model toward a correct or incorrect reasoning path. Specifically, when the reasoning trace "There are 67 fourth graders total. 38 are girls." is provided as input, the model still produces the correct final answer via greedy sampling. This shows that the "." token is not a decisive token.

**Branching Method:** To address this challenge, we propose a method based on *iterative greedy decoding with partial prefixes*. Suppose we have a correct reasoning trace (i.e. token sequence) $T^{\text{corr}} = (x_1, x_2, \ldots, x_n)$, obtained via greedy decoding, and an incorrect reasoning trace $T^{\text{incorr}} = (y_1, y_2, \ldots, y_m)$, obtained via non-greedy decoding. Our goal is to identify the pivotal token in $T^{\text{incorr}}$ that steers the model toward an incorrect final answer. Formally, let $f(\cdot)$ denote the final answer of a reasoning trace generated via greedy decoding, and let $\mathcal{A}_{\text{corr}}$ and $\mathcal{A}_{\text{incorr}}$ denote the correct and incorrect final answers, respectively. Then a token $y_k$ is defined as pivotal if $f(y_1, \ldots, y_{k-1}) \in \mathcal{A}_{\text{corr}}$ and $f(y_1, \ldots, y_k) \in \mathcal{A}_{\text{incorr}}$.

Operationally, we construct a prefix of length $k$ from $T^{\text{incorr}}$, i.e., $(y_1, \ldots, y_k)$, and feed it into the model to complete the reasoning trace using greedy decoding. We then check whether the resulting final answer is correct. If it is correct, we extend the prefix by adding the next token $y_{k+1}$ and repeat the procedure. If the final answer is incorrect, then the newly added token $y_k$ is identified as the pivotal token, since its inclusion causes greedy decoding to lead to an incorrect outcome. In the example shown in Figure 2, it is the "*no*" token which pushes the model trajectory towards an incorrect final answer. Hence, it becomes the pivotal token.

In the opposite case, when greedy decoding yields an incorrect reasoning trace while non-greedy decoding yields a correct one, we apply the same procedure. The difference is that the pivotal token is now defined as the first token in the non-greedy trace whose inclusion in the prefix causes greedy decoding to switch from an incorrect to a correct final answer. In this case, $y_k$ is pivotal if $f(y_1, \ldots, y_{k-1}) \in \mathcal{A}_{\text{incorr}}$ and $f(y_1, \ldots, y_k) \in \mathcal{A}_{\text{corr}}$.

Finally, after identifying the intervention token and corresponding pair of reasoning traces for a given GSM-Symbolic instance, we construct the *Error-Localization* dataset. As illustrated in Figure 3 each instance in this dataset consists of three components: 1) *Prefix*: the shared reasoning trace up to and

---

**Error-Localization Dataset Example**

**prefix:** There are 67 fourth graders in total. 38 are
girls, so 67 - 38 = 29 are boys. 6 girls were absent, but
**desired_token:** the
**undesired_token:** no

---

Figure 3: The Error-Localization dataset contains three components: *prefix*: the shared reasoning trace between the correct and incorrect paths (including intervention token), *desired_token*: the token the model should generate to produce the correct answer, and *undesired_token*: the token the model should avoid generating to ensure the correct answer.

including the intervention token, 2) *Desired token*: the token following the intervention token in the correct reasoning trace, and 3) *Undesired token*: the token following the intervention token in the incorrect reasoning trace.

### 3.1.2 IDENTIFYING CONSTRUCTIVE CIRCUITS WITH DCM

Using the training dataset generated in the previous step, we can localize errors in an incorrect reasoning trace to specific tokens. However, we can go further and identify the model components responsible for promoting the correct reasoning trace, or more specifically, the generation of the desired token, as shown in Figure 1(b). To achieve this, we leverage the Desiderata-based Component Masking (DCM) technique (Davies et al., 2023; De Cao et al., 2022; Prakash et al., 2024; 2025).

DCM learns a binary mask over key, query, and value weight matrices of all attention heads and MLP neurons in the LM by minimizing a tailored loss function. More specifically, it learns $n\_heads + 2 * n\_key\_value\_heads + n\_mlp\_neurons$ parameters for each layer, where $n\_heads$ represents the number of attention heads, $n\_key\_value\_heads$ represents the number of key and value heads in Grouped Attention, and $n\_mlp\_neurons$ represents the number of MLP neurons. Each parameter in the mask represents whether its corresponding model component should be intervened on or left unchanged during the forward pass. We use the following equation to update a model component's output using the mask:

$$h_{org} = m_i * 2 * h_{org} + (1 - m_i) * h_{org} \tag{1}$$

where $h_{org}$ represents the original model component output and $m_i$ represents the corresponding mask value. Concretely, if a component's mask value is 1, its activation is scaled by 2; otherwise, it remains unchanged. It is implemented using NNsight (Fiotto-Kaufman et al., 2024).

Since our goal is to isolate the components that promote the desired token while suppressing the undesired one, we use a loss function defined as the logit difference between the undesired and desired tokens to optimize the binary mask. To encourage sparsity in the mask, we add an $L_1$-norm regularization term, weighted by the hyperparameter $\lambda$. It ensures that only a small subset of components is identified as influential. Formally, the loss is:

$$\mathcal{L} = -(\text{logit}_{\text{desired\_token}} - \text{logit}_{\text{undesired\_token}}) + \lambda \sum \mathbf{m} \tag{2}$$

where $\lambda$ controls the sparsity of the binary mask, and its optimal value is selected by sweeping over a range of candidate values. We report the percentage of mask components in the results section, computed as $\frac{|M_{\text{learned}}|}{|M|}$, where $|M_{\text{learned}}|$ denotes the number of components selected by the learned mask and $|M|$ is the total number of components in the mask.

We train the binary mask using the Adam optimizer for 50 epochs, with a learning rate of 5e-3, batch size of 8, and tuning the $\lambda$ via parameter sweeps. Full details are in Appendix C. To prevent unnecessary computation, we apply early stopping: if the mask remains unchanged after 20% of batches in an epoch (i.e., the set of selected components does not vary), training is halted. Additionally, after each gradient update, we clamp the mask values to the range $[0, 1]$, as values outside this interval are incompatible with Equation 1.

In summary, after optimization, the learned mask identifies a small set of model components, *the circuit*, whose **amplified outputs** steer the model toward the correct reasoning trace and final answer.

### 3.1.3 TARGETED PARAMETER UPDATES

After identifying the model components that promote the desired token, we update only these components using gradient descent, as illustrated in Figure 1(c). We use the negative logit difference between the desired and undesired tokens as the loss function from the Error-Localization dataset. Gradients are applied exclusively to the previously identified components. Because the training dataset is small and only a limited number of updates are expected, we compute gradients over the entire dataset rather than using mini-batches.

We perform a total of 50 gradient update steps, evaluating the model's exact match accuracy on the validation set every 2 steps up to step 10, and subsequently every 10 steps. At the end of training, we select the best-performing updated model and evaluate it on the test set and report the results in Section 5. The optimal learning rate is determined through a sweep over candidate values[1].

## 4 EXPERIMENTAL SETUP

### 4.1 DATASETS FOR MATH AND GENERAL ABILITIES

We evaluate the effectiveness of CircuitTuning on improving the mathematical reasoning capabilities of LMs while preserving other skills gained during pretraining. For math reasoning, we use the GSM-Symbolic (Mirzadeh et al., 2025) benchmark, which provides templates derived from the GSM8K dataset (Cobbe et al., 2021). The benchmark contains 100 math problem templates across diverse topics, each with 50 instances. We randomly divide these instances into training, validation, and test sets in proportions of 0.52, 0.08, and 0.40, respectively. Thus, for each of the 100 templates, there are 26 training, 4 validation, and 20 test instances. We further filter our train split *to only include templates whose mean accuracy is below* 0.8 on the target model. A full list of selected templates is provided in Appendix B. In the rest of this paper, we refer to these splits as **GSym-Train**, **GSym-Val**, and **GSym-Test** respectively.

It is important to note that the model does not always produce a counterfactual reasoning trace under non-greedy sampling. Consequently, some GSM-Symbolic instances are absent from the Error-Localization dataset for the prefix and branching methods. As a result, the size of the training dataset varies across models and localization generation types, and it is always smaller than the maximum of 2600. Table 1 reports the training set sizes for all models considered. The validation and test sets consistently contain 400 and 2000 instances, respectively, from the original GSM-Symbolic dataset.

In addition to mathematical reasoning, we also evaluate the general capabilities of LMs using the MMLU, TriviaQA, and TruthfulQA benchmarks (Hendrycks et al., 2021; Joshi et al., 2017; Lin et al., 2022). MMLU includes questions spanning a broad range of topics. To better assess any unintended effects of enhancing math reasoning, a skill central to many STEM tasks, we evaluate on two MMLU subsets: "MMLU Stem" and "MMLU Humanities" as defined within MMLU. A complete list of both STEM and Humanities categories is provided in Appendix D.

### 4.2 EVALUATED MODEL FAMILIES

We evaluate CircuitTuning across multiple families of open-weight LLMs to assess its robustness and generality. We focus on the Gemma and OLMo model families (Team et al., 2024; Groeneveld et al., 2024). Specifically, we analyse `Gemma-2-9b-Instruct`, `Gemma-2-2b-Instruct`, `OLMo-2-1124-13B-Instruct`, and `OLMo-2-1124-7B-Instruct` models. Our manual inspection of erroneous reasoning traces of these models reveals that most errors stem from failures in logical reasoning steps rather than from arithmetic mistakes. For example, Figure 2 illustrates how `Gemma-2-9b-Instruct` produces an incorrect answer to a GSM-Symbolic instance due to its inability to extract the necessary information from the question, rather than an arithmetic error.

### 4.3 BASELINE: LORA FINE-TUNING

We compare our method against LoRA fine-tuning, a well-established parameter-efficient fine-tuning method (Hu et al., 2022) often used for task-adaptation. We use the same GSM-Symbolic

---

[1]Candidate learning rates: 1e-2, 5e-3, 1e-3, 5e-4, 1e-4, 5e-5, 1e-5.

| Configuration | Dataset | Dataset Size | % Mask | GSym-Test Acc | Std | Δ % Acc |
|---|---|---|---|---|---|---|
| **Gemma-2-9B-Instruct** | | | | | | |
| Original Model | – | – | – | 0.807 | – | – |
| CircuitTuning w mask | Prefix | 510 | 0.13% | 0.848 | ±0.006 | 4.1 |
| CircuitTuning w/o mask | Prefix | 510 | – | 0.849 | ±0.011 | 4.2 |
| CircuitTuning w mask | Branching | 512 | 0.17% | 0.881 | ±0.015 | **7.4** |
| CircuitTuning w/o mask | Branching | 512 | – | 0.875 | ±0.010 | 6.8 |
| LoRA Finetuning | **GSym-Train** | 676 | – | 0.850 | – | 4.3 |
| **Gemma-2-2B-Instruct** | | | | | | |
| Original Model | – | – | – | 0.411 | – | – |
| CircuitTuning w mask | Prefix | 1,283 | 0.92% | 0.440 | ±0.011 | 2.9 |
| CircuitTuning w/o mask | Prefix | 1,283 | – | 0.502 | ±0.018 | 9.1 |
| CircuitTuning w mask | Branching | 1,244 | 1.59% | 0.525 | ±0.010 | 11.4 |
| CircuitTuning w/o mask | Branching | 1,244 | – | 0.532 | ±0.009 | 12.1 |
| LoRA Finetuning | **GSym-Train** | 2,028 | – | 0.579 | – | **16.8** |
| **OLMo-2-1124-13B-Instruct** | | | | | | |
| Original Model | – | – | – | 0.742 | – | – |
| CircuitTuning w mask | Prefix | 864 | 0.37% | 0.768 | ±0.018 | 2.6 |
| CircuitTuning w/o mask | Prefix | 864 | – | 0.762 | ±0.002 | 2.0 |
| CircuitTuning w mask | Branching | 845 | 0.44% | 0.786 | ±0.005 | 4.4 |
| CircuitTuning w/o mask | Branching | 845 | – | 0.784 | ±0.006 | 4.2 |
| LoRA Finetuning | **GSym-Train** | 1,118 | – | 0.797 | – | **5.5** |
| **OLMo-2-1124-7B-Instruct** | | | | | | |
| Original Model | – | – | – | 0.739 | – | – |
| CircuitTuning w mask | Prefix | 974 | 0.19% | 0.772 | ±0.018 | 3.3 |
| CircuitTuning w/o mask | Prefix | 974 | – | 0.777 | ±0.006 | 3.8 |
| CircuitTuning w mask | Branching | 983 | 0.25% | 0.794 | ±0.006 | 5.5 |
| CircuitTuning w/o mask | Branching | 983 | – | 0.806 | ±0.012 | **6.7** |
| LoRA Finetuning | **GSym-Train** | 1,222 | – | 0.746 | – | 0.7 |

Table 1: Performance comparison of CircuitTuning and LoRA fine-tuning across multiple models on the GSM-Symbolic benchmark. For each model, we report accuracy on the **GSym-Test** under different training configurations: (i) CircuitTuning with a mask (updates restricted to components identified by the learned mask), (ii) CircuitTuning without a mask (updates applied more broadly), and (iii) LoRA fine-tuning. Results are shown for both Prefix- and Branching-based localization datasets. We also report dataset sizes, the percentage of model components updated, mean test accuracy with standard deviation, and the absolute accuracy improvement (Δ% Acc).

data splits used for CircuitTuning as described in 4.1. We apply LoRA to both the attention and MLP components of each transformer block, and run finetuning with an effective batch size of 32 for two epochs. We evaluate the model on the validation set every 10 steps with the same exact-match metric used for CircuitTuning and select the best-performing model checkpoint. We also sweep over a number of learning rates and report the best performing results on the test set (**GSym-Test**) in Section 5. Other LoRA hyperparameters, such as rank, learning rate schedule, etc, are fixed for all models and complete details can be found in E.1.

## 5 EXPERIMENTAL RESULTS ON MATH REASONING

This section presents the results of the original unmodified models, the models updated with Circuit-Tuning, and LoRA finetuned models on **GSym-Test** (CircuitTuning results are averaged over three

random seeds). For each localization dataset generation type, we report two configurations: (1) *CircuitTuning w/ mask*, where only the model components identified by the mask are updated, and (2) *CircuitTuning w/o mask*, an ablation where we skip model component localization via DCM and allow any model component to be updated during the gradient update step. The purpose of reporting both configurations is to disentangle the effects of model component localization and reasoning token localization.

Table 1 presents the results, highlighting a few key observations. First, models updated with CircuitTuning show substantially better performance than their corresponding base models across multiple model families. The improvement can be as high as 12.1% (for `Gemma-2-2b-Instruct`) using only 1244 samples. Moreover, CircuitTuning surpasses LoRA, a strong baseline, in both the `Gemma-2-9b-Instruct` and `OLMo-2-1124-7B-Instruct` models. These results indicate that CircuitTuning is effective for fine-tuning specific capabilities of LMs, particularly in data-constrained settings where maximizing performance from limited samples is critical.

Second, we find that models updated with the Branching localization dataset consistently outperform those updated with the Prefix method across model families. This suggests that accurately identifying the pivotal reasoning token that steers the model toward incorrect reasoning is critical for improving performance. More broadly, it underscores the importance of developing improved techniques for localizing token(s) within long reasoning traces, as a means of uncovering the underlying circuits and mechanisms responsible for different reasoning tasks.

Finally, we observe that the number of attention heads and MLP neuron weights that need to be updated to improve performance constitutes only a small fraction of the total model components. For example, achieving a 7.4% performance gain in `Gemma-2-9b-Instruct` requires updating only 0.17% of its components. This suggests that a limited subset of model components is primarily responsible for correct reasoning on the GSM-Symbolic dataset, and amplifying their contribution can significantly enhance performance. Moreover, since CircuitTuning updates only a small fraction of the model, we expect minimal interference with other capabilities acquired during pre-training, a hypothesis we evaluate in the following Section 6.

## 6 PRESERVING BROADER LM ABILITIES

Results in Table 1 show that CircuitTuning is effective in improving task performance for multiple models. However, its potential side effects on broader capabilities remain unclear. To address this, we evaluate the general abilities of updated LMs using widely adopted benchmarks, including MMLU, TriviaQA, and TruthfulQA (Hendrycks et al., 2021; Joshi et al., 2017; Lin et al., 2022) using LM-evaluation-harness (Gao et al., 2024). Specifically, we compare the best updated model with CircuitTuning against the original model as well as LoRA finetuned model to assess the impact of these updates on overall performance. As described in Section 4.1, we report two values for the MMLU benchmark: the mean accuracy over STEM topics and over Humanities topics.

As shown in Table 2, models updated with CircuitTuning achieve performance comparable to the base model on various standard benchmarks, suggesting that they retain most of the capabilities acquired during pretraining. The combination of targeted accuracy gains with minimal degradation highlights CircuitTuning as a safe and effective method for fine-tuning, particularly in applications where retaining broad competencies is essential.

## 7 DISCUSSION AND CONCLUSION

This work introduced CircuitTuning, a targeted model update method for amplifying specific model capabilities while preserving general performance. Unlike conventional finetuning strategies that update a large number of model components, CircuitTuning identifies pivotal reasoning errors and the circuit responsible for correct reasoning, then applies updates only to those components. Our experiments demonstrate that CircuitTuning substantially improves mathematical reasoning across multiple model families, up to +11.4% accuracy gain, while modifying as little as 1.59% of components. Importantly, these improvements come with minimal degradation on general-purpose benchmarks such as MMLU, TriviaQA, and TruthfulQA. These findings suggest that model capabilities

| Configuration | GSym-Test | MMLU Humanities | MMLU STEM | TriviaQA | TruthfulQA |
|---|---|---|---|---|---|
| **Gemma-2-9B-Instruct** | | | | | |
| Prefix w mask | 4.1 | 0.1 | 0.6 | -0.4 | 0.0 |
| Prefix w/o mask | 4.2 | -0.4 | 0.6 | -4.0 | 0.1 |
| Branching w mask | **7.4** | 0.2 | 0.8 | 2.0 | -0.3 |
| Branching w/o mask | 6.8 | 0.2 | 0.7 | 0.0 | -0.3 |
| LoRA | 4.3 | 0.3 | 0.5 | 1.9 | -0.1 |
| **Gemma-2-2B-Instruct** | | | | | |
| Prefix w mask | 2.9 | 0.2 | 0.1 | 0.8 | -0.7 |
| Prefix w/o mask | 9.1 | -0.6 | 0.8 | -1.3 | -1.6 |
| Branching w mask | 11.4 | 0.0 | 0.3 | 1.3 | 0.1 |
| Branching w/o mask | 12.1 | 0.3 | 0.6 | 1.6 | 0.7 |
| LoRA | **16.8** | -1.0 | 0.5 | 0.5 | -2.0 |
| **OLMo-2-1124-13B-Instruct** | | | | | |
| Prefix w mask | 2.6 | 0.1 | 0.1 | -0.4 | -0.1 |
| Prefix w/o mask | 2.0 | 0.4 | 0.0 | -0.0 | -0.3 |
| Branching w mask | 4.4 | 0.1 | 0.1 | -0.6 | -0.3 |
| Branching w/o mask | 4.2 | 0.0 | -0.2 | -0.5 | -0.5 |
| LoRA | **5.5** | 0.4 | 0.2 | 1.7 | 0.1 |
| **OLMo-2-1124-7B-Instruct** | | | | | |
| Prefix w mask | 3.3 | -0.5 | -0.4 | -0.0 | -0.0 |
| Prefix w/o mask | 3.8 | 0.1 | -0.3 | -0.3 | 0.8 |
| Branching w mask | 5.5 | -0.1 | -0.1 | -0.1 | 0.1 |
| Branching w/o mask | **6.7** | 0.4 | 0.2 | -0.6 | 2.0 |
| LoRA | 0.7 | -0.1 | -0.3 | 0.4 | -0.2 |

Table 2: Absolute percentage difference (in 0–100 scale) between original model and updated model for four CircuitTuning conditions and LoRA. Results are shown over five benchmarks: **GSym-Test**, MMLU Humanities, MMLU STEM, TriviaQA, and TruthfulQA.

are often governed by sparse, localized subnetworks that can be selectively strengthened to achieve reliable skill amplification.

Beyond improving mathematical reasoning, CircuitTuning highlights a broader principle: mechanistically informed, sparse updates provide a pathway to safe and effective model adaptation. This offers practical benefits for real-world deployment, where users expect improvements in targeted abilities without unexpected trade-offs, and contributes to the growing intersection between parameter-efficient finetuning and interpretability-guided model editing.

There are, however, limitations. We focused on mathematical reasoning as a testbed, leaving open whether similar gains can be achieved for other complex domains such as code generation or scientific problem solving. The current approach requires constructing error-localization datasets for token localization, which may be costly in settings without well-defined correctness signals. Further, our experiments considered only a single fine-tuning stage, whereas deployed models often undergo multiple rounds of fine-tuning to enhance different capabilities. However, in such continual learning settings, conventional techniques may introduce substantial regressions (Scialom et al., 2022; Luo et al., 2025), making CircuitTuning a promising alternative.

Future work could address these limitations by (i) extending CircuitTuning to other capabilities and domains, such as code generation, scientific reasoning, or multi-modal tasks, thereby testing its generality beyond symbolic math; (ii) automating the error-localization process using frontier LLMs to reduce reliance on multiple generations and improve scalability to datasets with longer reasoning traces; and (iii) incorporating more sophisticated optimization techniques to refine model components more efficiently than naive gradient descent.

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

## A  THE USE OF LARGE LANGUAGE MODELS (LLMs)

We used LLMs as a writing assistant to correct grammatical and typographical errors; beyond this, they did not contribute to any stage of the research.

## B  TEMPLATES UTILIZED IN ERROR-LOCALIZATION DATASET GENERATION

| Model | Template IDs |
|---|---|
| `gemma-2-9b-it` | 520, 364, 116, 184, 984, 1247, 480, 266, 20, 410, 43, 1207, 456, 989, 357, 1133, 1165, 434, 406, 1239, 858, 1088, 1021, 39, 652, 976 |
| `gemma-2-2b-it` | 520, 1305, 164, 991, 740, 103, 491, 364, 365, 496, 116, 125, 1025, 1031, 1020, 145, 401, 788, 918, 921, 158, 930, 1189, 1063, 184, 440, 458, 718, 728, 984, 473, 1247, 480, 636, 1277, 1026, 265, 266, 11, 20, 410, 1053, 800, 546, 937, 43, 1207, 1084, 320, 456, 982, 989, 99, 357, 1133, 1141, 737, 554, 1165, 1264, 304, 242, 434, 336, 661, 406, 1239, 858, 955, 1088, 459, 1021, 39, 74, 107, 652, 1164, 976 |
| `OLMo-2-1124-7B-Instruct` | 520, 1305, 991, 103, 873, 116, 1031, 1020, 145, 788, 184, 728, 984, 1247, 1275, 636, 1026, 265, 266, 11, 20, 410, 1053, 800, 43, 1207, 320, 989, 99, 357, 1133, 554, 1165, 1264, 304, 242, 434, 336, 406, 1239, 858, 1088, 459, 1021, 39, 652, 976 |
| `OLMo-2-1124-13B-Instruct` | 520, 1305, 300, 991, 740, 103, 364, 116, 184, 728, 984, 473, 1247, 1275, 1026, 265, 266, 11, 20, 410, 1053, 43, 1207, 1084, 320, 989, 99, 357, 1133, 554, 1264, 304, 434, 406, 1239, 858, 1088, 459, 39, 74, 107, 652, 976 |

Table 3: Template IDs used for training across different models.

## C  DCM TUNING DETAILS

| Category | Hyperparameter | Value |
|---|---|---|
| Training schedule | Base learning rate | 5e-3 |
| | Epochs | 50 |
| | Effective batch size | 8 |
| Optimization | Optimizer | Adam |
| | $\beta$ | 0.9 |
| | $\lambda$ | {1e-2, 1e-3, 5e-3, 1e-4} |

Table 4: Hyperparameters used for Desiderata-based Component Masking (DCM) tuning. Values include training schedule, optimizer settings, and $\lambda$ sweep for sparsity control.

## D  MMLU CATEGORIES

### D.1  MMLU STEM TASKS

mmlu_abstract_algebra, mmlu_anatomy, mmlu_astronomy, mmlu_college_biology, mmlu_college_chemistry, mmlu_college_computer_science, mmlu_college_mathematics,

mmlu_college_physics, mmlu_computer_security, mmlu_conceptual_physics, mmlu_electrical_engineering, mmlu_elementary_mathematics, mmlu_high_school_biology, mmlu_high_school_chemistry, mmlu_high_school_computer_science, mmlu_high_school_mathematics, mmlu_high_school_physics, mmlu_high_school_statistics, mmlu_machine_learning

## D.2 MMLU HUMANITIES TASKS

mmlu_formal_logic, mmlu_high_school_european_history, mmlu_high_school_us_history, mmlu_high_school_world_history, mmlu_international_law, mmlu_jurisprudence, mmlu_logical_fallacies, mmlu_moral_disputes, mmlu_moral_scenarios, mmlu_philosophy, mmlu_prehistory, mmlu_professional_law, mmlu_world_religions

## E TRAINING DETAILS

### E.1 LoRA FINETUNING DETAILS

| Category | Hyperparameter | Value |
|---|---|---|
| Training schedule | Base learning rate | {3e-5, 5e-5, 1e-4, 3e-4} |
| | Warmup steps | 5 |
| | Schedule | Linear warmup $\rightarrow$ cosine decay |
| | Epochs | 2 |
| | Effective batch size | 32 |
| Optimization | Optimizer | AdamW |
| | $(\beta_1, \beta_2)$ | 0.9, 0.999 |
| | $\epsilon$ | 1e-8 |
| | Max grad norm | 1.0 |
| Regularization | Weight decay | 0.01 |
| | Dropout | 0.1 |
| LoRA adapter | Rank ($r$) | 16 |
| | Alpha ($\alpha$) | 32 |

Table 5: LoRA fine-tuning hyperparameters. The exact number of optimization steps varies for each model based on the size of the training data.

Table 6: LoRA finetuning results across learning rates. The first row for each model is the unmodified (Original) model.

| Model | Learning Rate | GSym-Test | MMLU STEM | MMLU Humanities | Trivia QA | Truthful QA |
|---|---|---|---|---|---|---|
| | Original | 0.411 | 0.446 | 0.474 | 0.409 | 0.424 |
| | 3e-6 | 0.427 | 0.446 | 0.474 | 0.406 | 0.423 |
| | 5e-6 | 0.360 | 0.450 | 0.472 | 0.407 | 0.419 |
| | 1e-5 | 0.342 | 0.453 | 0.472 | 0.412 | 0.411 |
| Gemma2 2B | 2e-5 | 0.399 | 0.459 | 0.468 | 0.419 | 0.403 |
| | 3e-5 | 0.480 | 0.462 | 0.467 | 0.419 | 0.401 |
| | 5e-5 | **0.579** | 0.451 | 0.464 | 0.414 | 0.404 |
| | 1e-4 | 0.562 | 0.444 | 0.467 | 0.417 | 0.415 |
| | 3e-4 | 0.578 | 0.454 | 0.460 | 0.410 | 0.397 |
| | Original | 0.807 | 0.609 | 0.620 | 0.536 | 0.536 |
| | 3e-6 | 0.810 | 0.609 | 0.620 | 0.535 | 0.537 |
| | 5e-6 | 0.808 | 0.610 | 0.619 | 0.536 | 0.538 |
| | 1e-5 | 0.805 | 0.610 | 0.619 | 0.535 | 0.535 |
| Gemma2 9B | 2e-5 | 0.793 | 0.612 | 0.621 | 0.547 | 0.547 |
| | 3e-5 | **0.850** | 0.614 | 0.623 | 0.555 | 0.535 |
| | 5e-5 | 0.668 | 0.615 | 0.623 | 0.560 | 0.532 |
| | 1e-4 | 0.609 | 0.617 | 0.632 | 0.581 | 0.528 |
| | 3e-4 | 0.601 | 0.611 | 0.628 | 0.573 | 0.523 |
| | Original | 0.739 | 0.507 | 0.557 | 0.622 | 0.468 |
| | 3e-6 | 0.741 | 0.507 | 0.556 | 0.623 | 0.467 |
| | 5e-6 | 0.745 | 0.507 | 0.557 | 0.622 | 0.468 |
| | 1e-5 | **0.746** | 0.504 | 0.556 | 0.626 | 0.466 |
| Olmo 7B | 2e-5 | 0.644 | 0.505 | 0.557 | 0.630 | 0.465 |
| | 3e-5 | 0.660 | 0.507 | 0.556 | 0.630 | 0.465 |
| | 5e-5 | 0.585 | 0.509 | 0.559 | 0.630 | 0.463 |
| | 1e-4 | 0.616 | 0.514 | 0.559 | 0.634 | 0.461 |
| | 3e-4 | 0.442 | 0.514 | 0.559 | 0.638 | 0.447 |
| | Original | 0.742 | 0.564 | 0.601 | 0.703 | 0.512 |
| | 3e-6 | 0.754 | 0.565 | 0.602 | 0.705 | 0.512 |
| | 5e-6 | 0.772 | 0.565 | 0.603 | 0.712 | 0.512 |
| | 1e-5 | **0.797** | 0.566 | 0.605 | 0.720 | 0.513 |
| Olmo 13B | 2e-5 | 0.767 | 0.567 | 0.605 | 0.723 | 0.509 |
| | 3e-5 | 0.775 | 0.566 | 0.605 | 0.724 | 0.509 |
| | 5e-5 | 0.776 | 0.567 | 0.606 | 0.726 | 0.507 |
| | 1e-4 | 0.697 | 0.569 | 0.601 | 0.727 | 0.504 |
| | 3e-4 | 0.540 | 0.567 | 0.605 | 0.737 | 0.499 |

