# OpenReview forum: "CircuitTuning: Improving Math Reasoning in LLMs via Targeted Sub-Network Updates"
_ICLR.cc/2026/Conference — ICLR 2026 Conference Withdrawn Submission_

### Official Review · Reviewer_iuiG · 2025-10-28

**Soundness:** 2
**Presentation:** 2
**Contribution:** 2
**Rating:** 4
**Confidence:** 4

**Summary:**

This paper proposes a circuit-based parameter-efficient finetuning on reasoning tasks involving chain-of-thought style reasoning traces.
Specifically, the method consists of three steps:
1. Identifying a "pivot token" in the reasoning trace, which is a token at a position with a large impact on whether the reasoning trace will lead to the correct or wrong answer.
2. Identifying a circuit that is responsible for producing this pivot token, and by extension, for the model producing the correct or wrong answer.
3. Selectively finetuning only this circuit on the task of interest.
The paper shows that this method yields improvements generally comparable to or better than another parameter-efficient finetuning method, namely LoRA.

**Strengths:**

The first step of the proposed method, i.e., identifying a pivot token in chain-of-thought reasoning traces appears could be useful for interpretability research in general, since one limitation of all currently available, practical circuit analysis methods is that they require a minimal pair of inputs that differ in only a single token.

**Weaknesses:**

In my view, the contribution of this paper is not substantive enough.

From a novelty perspective, only the first step of the proposed method, i.e., the identification of "pivot tokens", is novel. Circuit identification (Step 2) is performed using an existing method, and selective finetuning (Step 3) has been proposed in prior work (Wang et al., ICLR 2025: HeadMap: Locating and Enhancing Knowledge Circuits in LLMs; https://openreview.net/forum?id=jUsrbOuQ5e).

Now, methodological novelty is not a necessary criterion, as there are many other kinds of contributions a paper can make. However, I'm struggling to identify what this other kind of contribution could be in the case of this paper. It's possible to see the paper as contributing an empirical comparison of the proposed parameter-efficient finetuning (PEFT) method to other PEFT methods, but this contribution is very limited for the following reasons:
- Evaluation is limited to only a subset of one benchmark, GSM-Symbolic. (Some more datasets appear in the paper, but these are used as control tasks to verify the absence of side effects, not for evaluating the efficacy of the proposed method)
- The comparison to existing PEFT methods is not systematically controlled: The number of finetuning instances is different and it is unclear if the comparisons are fair in terms of overall computational cost, since LoRA doesn't require circuit identification (which can be costly) but the proposed method does.
- Claimed parameter efficiencies ("modifying as little as 1.59% of model components") are potentially misleading, because "components" refers to both attention heads and MLP neurons. Since there are many more MLP neurons than heads, even finetuning all attention heads but no MLP neurons would likely yield a very low ratio of modified "model components" under this metric.

**Questions:**

Minor comment:

line 115: "We propose a novel technique, called CircuitTuning, to improve the mathematical reasoning capabilities of an LM, without affecting other abilities."
This is an imprecise statement since the experiments clearly show an impact on other abilities (Table 2). The above phrasing would be more appropriate if there were no statistically significant changes observed among all the tested abilities/benchmarks.

---

> ### Author Response · Authors · 2025-11-19
>
> Thank you for this thoughtful assessment. We agree that CircuitTuning builds on prior work in circuit identification and selective finetuning. However, the central contribution of our paper is to establish that the circuit analysis approach can be extended to reasoning settings to understand reasoning failures, which have a significantly larger number of tokens, in comparison to the narrow distribution tasks that have been analyzed in the existing mechanistic interpretability literature, to produce novel insights as well as improve model capabilities. Specifically:
>
>  - We introduce a new diagnostic framework that identifies where, in a reasoning trace, a model deviates despite having the latent ability to produce the correct answer.
>
>  - We show that the internal components causing this deviation can be isolated and directly corrected using a sparse, mechanistically grounded update.
>
>  - This constitutes the first demonstration that reasoning-trace deviations can be localized and causally repaired, rather than treated as black-box errors.
>
> Thus, our contribution lies in connecting token-level deviation detection → circuit-level diagnosis → targeted repair, yielding new insights into the structure of reasoning failures.
>
> We will make this scientific contribution more explicit in the revised manuscript.
>
> ---
>
> **line 115: "We propose a novel technique, called CircuitTuning...**
>
> Thank you for pointing it out. We will revise this sentence to “We propose a novel technique, called CircuitTuning, to improve the mathematical reasoning capabilities of an LM, with minimal impact on other abilities.”

---

> > ### Author Response · Authors · 2025-11-27
> >
> > We would like to thank the reviewer again for their feedback. We wanted to check whether our response addresses your concerns. We're happy to provide further clarification if needed.

---

### Official Review · Reviewer_U3ZQ · 2025-11-01

**Soundness:** 2
**Presentation:** 4
**Contribution:** 3
**Rating:** 4
**Confidence:** 4

**Summary:**

The paper introduces CircuitTuning, a novel, mechanistically-inspired method for improving the mathematical reasoning abilities of LLMs. The method operates in three stages: 1) It first generates both correct and incorrect reasoning traces for a given problem to identify the "pivotal token" where the model's reasoning diverges towards an error. 2) It then uses a masking technique (Desiderata-based Component Masking, DCM) to localize the specific attention heads and MLP neurons that are most responsible for generating the correct next token. 3) Finally, it applies targeted gradient updates exclusively to this sparse sub-network. The authors evaluate CircuitTuning on the GSM-Symbolic benchmark across four models from the Gemma and OLMo families. They show that their method can improve math reasoning accuracy by up to +12.1% while modifying a very small fraction of model parameters (as low as 0.17%), and importantly, without significantly degrading performance on general benchmarks like MMLU, TriviaQA, and TruthfulQA.

**Strengths:**

1.  The three-stage approach of localizing the error token, identifying the responsible components, and performing targeted updates is a novel way to bridge mechanistic interpretability and model fine-tuning. The "Branching Method" for finding the pivotal token is a particularly strong and well-motivated part of this methodology.

2.  The method's main strength is its ability to achieve significant performance improvements while modifying a tiny fraction of the model's parameters (e.g., 0.17% for Gemma-9B). The results in Table 2 convincingly show that this surgical approach avoids the catastrophic forgetting that can plague broader fine-tuning methods, preserving performance on general benchmarks.

3.   The paper is written with exceptional clarity. The method, experimental setup, and results are described in sufficient detail to facilitate understanding.

**Weaknesses:**

1.  The LoRA baseline is trained on a different (and often larger) dataset than CircuitTuning. The performance difference could be attributed to the curated, high-signal "Error-Localization" dataset rather than the targeted update mechanism itself. To isolate the benefit of the proposed update strategy, LoRA should be trained on the exact same dataset.

2.  The method's performance relative to LoRA is not consistent across models. LoRA significantly outperforms CircuitTuning on the Gemma-2B model and also wins on the OLMo-13B model. The paper lacks a discussion or analysis of why this might be the case.

3.  The method's reliance on generating paired correct/incorrect reasoning traces may be difficult to scale to more complex, open-ended domains like code generation or scientific reasoning, where a single, easily verifiable "correct trace" may not exist.

4.  In several cases (e.g., Gemma-2B Branching, OLMo-7B Branching), the "w/o mask" ablation performs better than the "w/ mask" version. This is counter-intuitive to the central hypothesis that updating only a sparse, localized circuit is optimal.

**Questions:**

1.  Could you please clarify the rationale for training the LoRA baseline on the larger GSym-Train set instead of the smaller, curated Error-Localization dataset used for CircuitTuning? To make a more compelling case for your method's targeted update mechanism, would it be possible to run an experiment where LoRA is trained on the exact same data used by CircuitTuning?

2.  What is your hypothesis for the inconsistent results when comparing CircuitTuning to LoRA? Specifically, why do you think LoRA achieves a much larger accuracy gain on Gemma-2B (+16.8%) and a better gain on OLMo-13B (+5.5%) compared to your method? Does the effectiveness of CircuitTuning depend on model scale or architecture?

3.  The "w/o mask" ablation, which performs token localization but allows gradient updates to all parameters, sometimes outperforms the "w/ mask" version. How do you interpret this result? Does it challenge the core assumption that only a very sparse set of components should be updated?

4.  How do you envision adapting CircuitTuning to tasks where reasoning errors are more semantic or distributed across a sequence, rather than hinging on a single pivotal token (e.g., improving factual consistency in a summary or stylistic tone in a story)?

---

> ### Author Response · Authors · 2025-11-19
>
> **The LoRA baseline is trained on a different (and often larger) dataset than CircuitTuning...**
>
> Thank you for raising this important point. We chose not to train LoRA on the Error-Localization dataset for two main reasons.
>
> 1. The localization dataset is an integral component of CircuitTuning. Its construction is part of the method itself, and our goal in the experiments was to compare the full CircuitTuning pipeline against standard baselines. Training LoRA on the Error-Localization dataset would amount to partially adopting elements of CircuitTuning rather than providing a clean comparison to existing finetuning approaches.
> 2. The localization dataset is substantially smaller than the dataset used for LoRA. Using such a small dataset for LoRA would significantly degrade its performance and risk presenting an artificially handicapped baseline. To ensure a fair and representative comparison, we trained LoRA on the standard, larger GSM-Symbolic training split, reflecting how LoRA is typically used in practice.
>
> We will clarify this reasoning in the revised manuscript.
>
> ---
>
> **The method's performance relative to LoRA is not consistent across models...**
>
> We acknowledge that CircuitTuning does not uniformly outperform LoRA on every model. However, we would like to emphasize that the primary contribution of this work is scientific understanding of reasoning failures, not solely benchmarking gains. CircuitTuning explicitly identifies where the model’s reasoning trace diverges despite possessing the latent ability to solve the problem and then isolates the internal components responsible for this deviation. These mechanistic insights directly guide the targeted parameter updates that follow.
>
> Because the method aims to illuminate why models fail on specific reasoning steps, and demonstrate that such failures can be corrected through sparse, mechanism-aligned updates, our goal is complementary to, rather than purely competitive with, LoRA-style finetuning. We will add a discussion of these factors and further contextualize the comparison in the revised version.
>
> ---
>
> **The method's reliance on generating paired correct/incorrect reasoning traces may be difficult to scale...**
>
> Thank you for raising this important point. We agree that the reliance on paired correct/incorrect reasoning traces presents a limitation when scaling to open-ended domains such as code generation or scientific reasoning, where a single verifiable “correct trace’’ may be unavailable. However, the central goal of this work is to deepen scientific understanding of reasoning errors by precisely identifying where a model deviates from a correct latent trajectory. Doing so necessarily requires access to verifiable correctness signals.
>
> We also note that this limitation is not unique to CircuitTuning, many post-training approaches similarly depend on ground-truth labels or an external verifier. For example, verifier-based fine-tuning and supervised reasoning methods such as RFT (Reinforced Fine-Tuning), Verifier-LLM, and GoT (Guided by Tests) all require explicit correctness judgments or programmatic checkers to supervise the learning signal. CircuitTuning fits within this broader landscape of methods that leverage correctness supervision to improve reasoning quality.
>
> We will clarify this limitation in the revised manuscript and discuss how future work may relax this requirement, such as using weak verifiers or probabilistic correctness signals in more open-ended settings.
>
> ---
>
> **In several cases (e.g., Gemma-2B Branching, OLMo-7B Branching), the "w/o mask" ablation performs better than the "w/ mask"...**
>
> Thank you for raising this important point. We would like to clarify that our claim is not that updating the localized circuit is always optimal in terms of raw performance. Instead, our hypothesis is that updating a localized subnetwork can achieve performance comparable to broader finetuning approaches while substantially reducing interference with other task mechanisms. In this sense, the goal is not to maximize accuracy at all costs, but to demonstrate that targeted, mechanism-aligned updates can correct specific reasoning failures without the collateral effects often observed when updating large portions of the model.
>
> The cases where the “w/o mask’’ ablation performs slightly better can be understood as scenarios where broader gradient flow provides more opportunities for improvement, but at the cost of reduced interpretability and potentially greater risk of overwriting unrelated capabilities. We will clarify this distinction in the revised manuscript and emphasize that the benefit of CircuitTuning lies in controlled, interpretable, and minimally disruptive updates rather than purely optimizing raw performance.

---

> > ### Author Response · Authors · 2025-11-19
> >
> > **How do you envision adapting CircuitTuning to tasks where reasoning errors are more semantic...**
> >
> > Thank you for this intriguing question. We believe CircuitTuning can be extended to settings where errors are distributed across multiple tokens rather than hinging on a single pivotal token. The high-level pipeline (1. identify token(s), 2. localize components, 3. perform targeted updates) remains the same; only the token-localization stage needs to be adapted. Concretely, one can apply any of the following strategies to produce a subset of pivotal tokens:
> >
> > 1. Generalized branching / prefix scan: extend the branching procedure to sliding windows of tokens (rather than single-token additions) and mark those windows whose inclusion flips the final outcome.
> >
> > 2. Attribution / influence methods: use gradient-based attributions, attention attribution, or influence function proxies to highlight sequence positions that most affect the undesired outcome.
> >
> > 3. Frontier-model proposals (contrastive pairs): use a stronger “frontier” LLM to propose a candidate set of pivotal tokens for each contrastive instance (i.e., ask the frontier model which tokens/phrases change the answer). This avoids exhaustive forward-pass searches in long traces.
> >
> > Once a subset of pivotal tokens or spans is identified, the DCM-based component-localization and the subsequent targeted-parameter-update steps proceed unchanged: we construct an error-localization dataset over the selected token subset, learn a sparse mask that promotes the desired token(s)/span(s), and apply sparse updates only to the identified components.
> >
> > We will add a short discussion and sketch of these extensions in the revision to clarify how CircuitTuning can be adapted to more semantic, distributed error modes.

---

> > > ### Author Response · Authors · 2025-11-27
> > >
> > > We would like to thank the reviewer again for their feedback. We wanted to check whether our response addresses your concerns. We're happy to provide further clarification if needed.

---

### Official Review · Reviewer_Bsk9 · 2025-11-01

**Soundness:** 1
**Presentation:** 2
**Contribution:** 1
**Rating:** 2
**Confidence:** 3

**Summary:**

This paper proposes a select-the-finetune method, that targets subnetworks important to reasoning during fine-tuning. The authors evaluated the method on several common benchmarks, and showed that the method outperforms LoRA under certain settings.

After reading the paper, I believe that although the proposed method is well grounded on previous related works, the overall framework is premature and computationally expensive, and the advantages of the method are not well presented in the paper. In addition, the evaluation setup is not well-designed, and the performance of the method is not consistent. Therefore I think major revision is needed for the current manuscript, and I recommend rejecting the paper.

**Strengths:**

1. The research problem in the paper is well-motivated, focusing on the circuit phenomenon in reasoning models.
2. The proposed method is clearly explained, combining existing diagnostic frameworks and reasoning-focused training.

**Weaknesses:**

Methodological:

1. The scalability of the proposed method is under question: procedures such as token localization could incur significantly larger cost as the model is scaled-up, since more challenging reasoning tokens/traces will need to be detected and selected.
2. The advantages of the proposed method is not clearly stated in the paper. Is it memory efficiency? Or better interpretability of the reasoning paradigm? Without stating the advantages, the method will likely be less recognized by practitioners or researchers.
3. The method seems too tailored in the sense that different models/settings may need significantly different training setups, which is not generalizable.

Experimental:

1. The performance improvement is not consistently better than other baselines, and the chosen baseline (LoRA) are not representative of the state-of-the-art. The authors should at least compare with more representative baselines, such as full fine-tuning or advanced LoRA methods [1]
2. The fairness of baseline comparison is under question: The authors did not explicitly compare the computational overhead of different methods. Since sub-procedures like token localization can incur substantial computational cost, the authors should explicitly mention it in the paper for clarification.
3. No supplementary materials or code is provided. This makes the reproducibility of the method under question.

[1] DoRA: Weight-Decomposed Low-Rank Adaptation, https://arxiv.org/abs/2402.09353

**Questions:**

1. Could the authors provide a more detailed explanation of the targeted parameter update procedure? It seems that the mask is fixed during training, which is counterintuitive, since we would think that the circuits should be dynamically changing.
2. There are previous work discussing subnetworks in reasoning model training, and sparse training on principal weights crucial to reasoning [1, 2]. It would be great if the authors could add discussions on these works in the paper.
3. The detailed computational overhead is not outlined in the paper. Could the author provide some detail in the cost of each stage of the proposed method, and compare with the baseline? This will ensure fairness of comparison.

[1] Reinforcement Learning Finetunes Small Subnetworks in Large Language Models, https://arxiv.org/abs/2505.11711

[2] LIFT the Veil for the Truth: Principal Weights Emerge after Rank Reduction for Reasoning-Focused Supervised Fine-Tuning, https://arxiv.org/abs/2506.00772

---

> ### Author Response · Authors · 2025-11-19
>
> **The scalability of the proposed method is under question...**
>
> We agree that token-level localization introduces additional computation, and we would like to clarify that this cost is not a flaw of the method but an inherent tradeoff between computational expenditure and the targetedness of the intervention. CircuitTuning is designed to sit at a specific point on this spectrum:
>  - More localization compute → more precise identification of pivotal tokens → higher-quality circuit selection → smaller, safer parameter updates.
>
>  - Less localization compute → cheaper preprocessing but coarser intervention → greater risk of modifying unrelated mechanisms.
>
> Our branching-based procedure intentionally invests modest additional compute to obtain highly targeted intervention points, which is precisely what enables us to update only 0.13%–1.59% of components while preserving performance on MMLU, TriviaQA, and TruthfulQA.
>
> Importantly, this tradeoff is controllable:
>  - Users can choose the cheaper prefix method when compute is constrained.
>  - Conversely, in settings where preserving broad capabilities is essential, spending slightly more localization compute yields substantially more precise updates.
>
> ---
>
> **The advantages of the proposed method is not clearly stated in the paper...**
>
> Thank you for the suggestion. We will clarify that the main contribution of our work is to show that failures in multi-step reasoning can be localized to specific circuits, and that these failure circuits can, in fact, be corrected through targeted model updates. Prior mechanistic interpretability work has largely focused on non-reasoning tasks with single-step outputs, where identifying the responsible components is more straightforward.
>
> By localizing failure circuits directly from reasoning traces and updating only those components, our method demonstrates that the circuit analysis approach can be extended to reasoning and can also guide sparse, targeted parameter changes that improve reasoning ability while minimizing interference with other skills. We will highlight these advantages more clearly in the revised manuscript.
>
> ---
>
> **The method seems too tailored...**
>
> Thank you for raising this concern. We respectfully disagree that CircuitTuning requires model-specific tailoring. The method is designed as a modular, plug-and-play framework: token localization, component localization, and sparse updates operate independently of the underlying architecture and can be applied to any model or task that provides reasoning traces. While, as with any training procedure, certain hyperparameters (e.g., learning rate or sparsity weight) may need tuning, the overall algorithmic pipeline remains unchanged across settings. We will clarify this generality in the revision.
>
> ---
>
> **The performance improvement is not consistently better than other baselines...**
>
> Thank you for this helpful suggestion. We would like to clarify that we did experiment with full finetuning, and the results were substantially worse than both CircuitTuning and LoRA across all models. We believe this is due to severe overfitting in the small-data regime we study, where updating all parameters leads to degradation rather than improvement.
>
> We also thank the reviewer for pointing us to DoRA [1]. We agree it is an important baseline and will include a comparison with DoRA in the next version of the paper.
>
> ---
>
> **The fairness of baseline comparison is under question...**
>
> Thank you for pointing this out. We agree that reporting the computational overhead of each method is important for fairness and clarity. We will update the paper to provide explicit details on the compute cost associated with token localization, DCM optimization, and the subsequent sparse updates, and we will compare these costs to those of LoRA and full finetuning in the revised version.
>
> ---
>
> **No supplementary materials or code is provided...**
>
> We will release the code with the next version of the paper.

---

> > ### Author Response · Authors · 2025-11-19
> >
> > **Could the authors provide a more detailed explanation of the targeted parameter update procedure...**
> >
> > Thank you for this question. We would like to clarify that the mask is not fixed during the model component–localization phase. During this phase, we use Desiderata-based Component Masking (DCM) to learn the mask by optimizing it to maximize the logit difference between the desired and undesired tokens derived from the error-localization dataset. Concretely, DCM assigns a learnable scalar to every attention head (K, Q, and V) and MLP neuron, and these mask values are updated via gradient descent across approximately 50 epochs. The optimization minimizes the loss
> >
> > $$L=−(logit_{desired}−logit_{undesired})+λ∑m$$
> >
> > where the first term encourages components that promote the correct token, and the L1 penalty enforces sparsity. Mask values are continuously updated, clamped to [0,1], and early stopping is triggered when the selected set of components stabilizes. Thus, the circuit is actively learned and not predetermined.
> >
> > Only after this localization step converges do we fix the mask. In the subsequent targeted parameter update phase, we update parameters only for the components selected by the learned mask. In this phase, the goal is not to continue discovering the circuit but to strengthen the already-identified constructive components. The loss used here is again the negative logit difference between desired and undesired tokens, but gradients flow only through masked components. This separation of (i) learning the circuit and (ii) updating that circuit is deliberate: updating both simultaneously risks destabilizing the mask and preventing it from converging to a sparse, interpretable set of components.
> >
> > ---
> >
> > **There are previous work discussing subnetworks in reasoning model training...**
> >
> > Thank you for pointing us to these relevant works. We agree that both [1, 2] are closely related to our focus on sparse, mechanism-level reasoning improvements. We will incorporate a discussion of these papers into the related works section of the revised version to better contextualize CircuitTuning within this emerging line of research.

---

> > > ### Author Response · Authors · 2025-11-27
> > >
> > > We would like to thank the reviewer again for their feedback. We wanted to check whether our response addresses your concerns. We're happy to provide further clarification if needed.

---

### Note · Authors · 2026-01-06

I have read and agree with the venue's withdrawal policy on behalf of myself and my co-authors.